# MESH-INDEPENDENT OPERATOR LEARNING FOR PDEs USING SET REPRESENTATIONS

## ABSTRACT

Operator learning, which learns the mapping between infinite-dimensional function spaces, is an attractive alternative to traditional numerical methods for solving partial differential equations (PDEs). In practice, the functions of the physical systems are often observed by sparse or even irregularly distributed measurements; thus, the functions are discretized and usually represented by finite structured arrays, which are given as data of input-output pairs. Through training with the arrays, the solution of the trained models should be independent of the discretization of the input function and can be queried at any point continuously. Therefore, the architectures for operator learning should be flexibly compatible with arbitrary sizes and locations of the measurements, otherwise, scalability can be restricted when the observations have discrepancies between measurement formats. In this study, we propose the proper treatment of discretized functions as set-valued data and construct an attention-based model, called mesh-independent operator learner (MIOL), to provide proper treatments of input functions and query coordinates for the solution functions by detaching the dependence of the input and output meshes. Our models pre-trained with benchmark datasets of operator learning are evaluated by downstream tasks to demonstrate the generalization abilities to varying discretization formats of the system, which are natural characteristics of the continuous solution of the PDEs.

## 1 INTRODUCTION

Partial Differential equations (PDEs) are among the most successful mathematical tools for representing the physical systems with governing equations over infinitesimal segments of the domain of interest, given some problem-specific boundary conditions or forcing functions (Mizohata, 1973). The governing PDEs, which are globally shared in the entire domain, are interpreted as interactions between infinitesimal segments with respect to their geometrical structures and values. Because of the universality of the entire domain, the system can be analyzed in a continuous manner with respect to the system inputs and outputs. In general, identifying appropriate governing equations for unknown systems is very challenging without domain expertise, however, numerous unknown processes remain for many complex systems. Even if knowing the governing equation of the system is known, it requires unnecessary time and memory costs to be solved using conventional numerical methods, and sometimes it is intractable to compute in a complex and large-scale system.

**Motivation.** In recent years, operator learning, an alternative to conventional numerical methods, has been gaining attention, pursuing mapping between infinite-dimensional input/output function spaces in a data-driven manner without any problem-specific knowledge of the system (Nelsen & Stuart, 2021; Li et al., 2020a;b; 2021b; Lu et al., 2019; 2021; Cao, 2021; Kovachki et al., 2021). Intuitively, for the underlying PDE, $\mathcal{L}_a u = f$ defined on the continuous bounded domains $\Omega$ with system parameters $a \in \mathcal{A}$, forcing function $f \in \mathcal{F}$, and the solution of the system $u \in \mathcal{U}$, the goal of the operator learning is to approximate the inverse operator $\mathcal{G} = \mathcal{L}_a^{-1} f : \mathcal{A} \to \mathcal{U}$ or $\mathcal{G} : \mathcal{F} \to \mathcal{U}$ with parametric model $\mathcal{G}_\theta$. Without loss of generality, when the input function is $a$, the output function can be computed as $u = \mathcal{G}_\theta(a)$. Because the operator $\mathcal{G}_\theta$ should be able to capture interactions between elements of system inputs $a$ to discover the governing PDEs, $\mathcal{G}_\theta$ is approximated by a series of integral operators with parameterized kernels that iteratively update the system input to the output (Nelsen & Stuart, 2021; Li et al., 2020a;b; 2021b; Cao, 2021; Kovachki et al., 2021). In practice,

Table 1: Comparison with other studies.

| | Operator networks (Lu et al., 2019; 2021) | Neural operators (Li et al., 2020a;b; 2021b) | MIOL (ours) |
|---|---|---|---|
| Decoupling input/output mesh | Yes | No | Yes |
| Mesh-independence to input | No, require fixed input mesh | Yes | Yes |
| Mesh-independence to output | Yes | No, depend on input mesh | Yes |

because continuous measurements of the input/output functions are infeasible, the observed data are provided as a set of input-output pairs, which are point-wise finite discretization of the functions.

The output values at coordinate $y$ can be expressed as $u(y) = [\mathcal{G}_\theta(a)](y)$ which can be viewed as the operating $\mathcal{G} : \mathcal{A} \times Y \to \mathcal{U}$ with two input placeholders, $a \in \mathcal{A}$ and $y \in Y$. Then, the following two considerations can be considered for the model with respect to the input function $a$, and query coordinate $y$: (1) the output of the model should not depend on any discretization format of $a$, and (2) the model should be able to output a solution at any query coordinate $y$. The measurements of the system are often sparsely and irregularly distributed owing to the geometry of the domain, environmental conditions, or inoperative equipment (Belbute-Peres et al., 2020; Lienen & Günnemann, 2022). In addition, in popular numerical methods for solving the PDEs, such as finite element methods, unstructured meshes are often utilized for the discretization of the domain, and adaptive remeshing schemes are commonly deployed where the regions of the domain require different resolutions depending on the accuracy of the prediction (Brenner et al., 2008; Huang & Russell, 2010). Thus, the model should aggregate global information over the measurements to process $[\mathcal{G}_\theta(a)]$ which can be reused for any discretization of $a$, and the solution $[\mathcal{G}_\theta(a)](y)$ should be queried at any coordinate $y$, such as implicit neural representations (Sitzmann et al., 2020).

**Existing architectures.** There are two representative frameworks for operator learning: operator networks, and neural operators. Based on the framework of operator networks (Chen & Chen, 1995), (Lu et al., 2019) extended the architectures with modern deep networks, called deep operator networks (DeepONets), which is a successful architectures for operator learning. To process the input functions and query coordinates, DeepONets consist of two sub-networks, called branch network and trunk network, respectively. DeepONets can be queried at any coordinate $y$ from the trunk network, however, they used the fixed discretization of the system inputs $a$ from the branch network (Lu et al., 2019; 2021). Another promising framework is neural operators, which consists of several integral operators with parameterized kernels to map between infinite-dimensional functions. Neural operators can be adapted to different resolutions of system inputs $a$ in a mesh-invariant manner (Li et al., 2020a;b). However, the implemented architectures of the neural operators are typically assumed to have the same sampling points for input and output functions (Li et al., 2020a;b; 2021b; Kovachki et al., 2021; Lu et al., 2022), that is, they have not been formulated as a method for decoupling the discretization of the system input and solution, leading to the solution of the neural operators not being flexibly queried at any coordinate $y$. In addition, widely used as successful model for operator learning due to their efficacy and accuracy (Li et al., 2021b; Pathak et al., 2022), Fourier neural operators (FNO) are limited to uniform grid discretization owing to the usage of FFT. The limitations and schematics of the existing studies are presented in Table 1 and Figure 6.

**Contributions.** There has been limited discussion on the generalization abilities of discretizations with extended variations, such as different sampling points for inputs and outputs functions with arbitrary numbers and irregular distributions. For these considerations, we treated the observational data as a set without intrinsic assumptions about the data and constructed what we call a mesh-independent operator learner (MIOL), as shown in Figure 1. MIOL is a fully attentional architecture consisting of an encoder-processor-decoder, where the encoder and decoder are made in a way related to set encoder-decoder frameworks (Zaheer et al., 2017; Lee et al., 2019) to detach the dependence of input and output meshes from the processor that processes the smaller fixed number of vectors in latent space. Attention mechanisms have been discussed as not only efficient in expressing pair-wise interactions (Cao, 2021; Kovachki et al., 2021; Guibas et al., 2021; Pathak et al., 2022) but also flexible in processing data in a modality-agnostic way (Vaswani et al., 2017; Jaegle et al., 2021; 2022). Finally, we conducted several experiments on the benchmark PDE datasets for operator learning (Li et al., 2021b). The results show that our model is not only competitive in original benchmark tasks but also robustly applicable when discretizations of input and output functions are allowed to be different, unstructured, and irregularly distributed, which are natural consequences of continuous solutions for physical systems but are not flexibly compatible with existing ones.

## 2 PROBLEM DEFINITION

Let the observed data be provided as $N$ instances of input-output pairs, $\{(a_i, u_i)\}_{i=1}^N$. $a_i = a_i|_X$ and $u_i = u_i|_Y$ are finite discretizations of the input function $a_i \in \mathcal{A}(\Omega_x; \mathbb{R}^{d_a})$ and output function $u_i \in \mathcal{U}(\Omega_y; \mathbb{R}^{d_u})$, where $\mathcal{A}$ and $\mathcal{U}$ are separable Banach spaces. $X = \{x_1, ..., x_{n_x}\}$ and $Y = \{y_1, ..., y_{n_y}\}$ are the finite sets of discretized points for the continuous bounded domains $\Omega_x$ and $\Omega_y$ with the number of discretized points $n_x$ and $n_y$, respectively. The training procedure of the original operator learning involves optimizing the following objective to learn a model $\mathcal{G}_\theta : \mathcal{A} \to \mathcal{U}$,

$$E_{a \sim \mu}\left[\mathcal{L}\left(\mathcal{G}_\theta(a), u\right)\right] = \int_{\mathcal{A}} \|u - \mathcal{G}_\theta(a)\|^2 d\mu(a) \approx \frac{1}{N}\sum_{i=1}^N \|u_i - \mathcal{G}_\theta(a_i)\|^2, \tag{1}$$

where $a \sim \mu$ is i.i.d on $\mathcal{A}$, and the discretizations of the input and output space are usually assumed to be the same (Li et al., 2020a;b; 2021b; Kovachki et al., 2021; Lu et al., 2022), that is, $X = Y$. In this study, we consider the extended situations where the discretized points $X$ and $Y$ are allowed to be different, that is, $X \neq Y$, and arbitrarily on the domain, while varying the number of $n_x$ and $n_y$ for each $i$. In practice, we aimed to build a model that is applicable to an irregularly distributed and varying number of discretization points during testing. Therefore, the goal is to learn the model $\mathcal{G}_\theta : \mathcal{A} \times Y \to \mathcal{U}$ that is expected to make the following test errors as small as possible when training with the above objective 1,

$$E_{a \sim \mu}E_{X,Y}\left[\mathcal{L}\left([\mathcal{G}_\theta(a)](y), u(y)\right)\right] = E_{a \sim \mu}\int_{\Omega_x}\int_{\Omega_y} \|u(y) - [\mathcal{G}_\theta(a|_X)](y)\|^2 dy d\nu(X), \tag{2}$$

where $X$ and $Y$ can be arbitrary discretizations of the input and output domains. Described in section 5.2, to evaluate generalization ability respect to varying the size of inputs, instead of calculating the test error 2, we calculate respective empirical test errors correspond to ratio $p$,

$$\frac{1}{N}\sum_{i=1}^N \|u_i(Y) - [\mathcal{G}_\theta(a_i^{(p)})](Y)\|^2. \tag{3}$$

where $X^{(p)}$ is a randomly masked discretization with a masking ratio $p$ of the given input domain $X$, $a_i^{(p)} = a_i|_{X^{(p)}}$ is $a_i$ evaluated on $X^{(p)}$, and the model is evaluated at all given discretized points of $Y$ (using all of $Y$ is not necessary). The empirical test error can be bounded by the sum of the approximation error and discretization errors from the discrepancy of the discretizations of the input function,

$$\|u_i(Y) - \mathcal{G}_\theta(a_i^{(p)})](Y)\| \leq \|u_i - \mathcal{G}_\theta(a_i)\| + \|[\mathcal{G}_\theta(a_i)](Y) - [\mathcal{G}_\theta(a_i^{(p)})](Y)\|. \tag{4}$$

The approximation error is expected to be sufficiently small by the training procedure 1, and if we consider the input functions as set representation, that is, $a_i^{(p)} \subset a_i$, there exists an appropriate masking ratio $p$, where the discretization errors can be expected to be sufficiently small by processing them with an efficient permutation-invariant set encoder whose outputs are independent of cardinalities and locations of input discretizations (Zaheer et al., 2017; Wagstaff et al., 2019; 2021).

## 3 PRELIMINARIES

### 3.1 NEURAL OPERATORS

The architectures of neural operator (Li et al., 2020a), usually consist of lifting-iterative updates-projection, which are corresponding to encoder-processor-decoder, respectively. The lifting $v_1(x) = \mathcal{P}(a(x))$ and projection $u(x) = \mathcal{Q}(v_L(x))$ are local transformations usually implemented by point-wise feed-forward neural networks for mapping input features to target dimensional features. The iterative updates $\mathcal{G}_l : v_l \mapsto v_{l+1}, l \in [1, L-1]$ are global transformations implemented by sequence of following transformations to capture the interactions between the elements,

$$v_{l+1}(x) = [\mathcal{G}_l(v_l)](x) = \sigma\left(W_l v_l(x) + [\mathcal{K}_l(v_l)](x)\right), \quad x \in \Omega, \tag{5}$$

where $\sigma$ are nonlinear functions, $W_l$ are point-wise linear transformations, $\mathcal{K}_l$ are kernel integral operations on $v_l(x)$. While the spatial domains of input and output are usually the same, the discretizations of input and output functions can be different with varying numbers and locations, $X \neq Y$. However, the existing implementations of the neural operators typically use the same discretizations for the input and output (Li et al., 2020a;b; 2021b; Kovachki et al., 2021; Lu et al., 2022), which leads to the predicted outputs only queried at the input meshes. Although the compatibility and performance of neural operators are mesh-invariant to inputs (stably low test errors are observed at different resolutions from training (Li et al., 2020a;b; 2021b)), the outputs cannot be flexibly queried to any points of coordinate $y$. For these reasons, separative treatments for input functions $a$ and query coordinates $y$ are required for decoupling the discretizations of inputs and outputs.

## 3.2 KERNEL INTEGRAL OPERATION AND ATTENTION

The kernel integral operations are generally implemented by integration of input values weighted by kernel values $\kappa$ representing the pair-wise interactions between the elements on input domain $x \in \Omega_x$ and output domain $y \in \Omega_y$,

$$[\mathcal{K}(v)](y) = \int_{\Omega_x} \kappa_\phi(y, x)v(x)dx, \quad (x, y) \in \Omega_x \times \Omega_y, \tag{6}$$

where the parameterized kernels $\kappa_\phi$ are defined on $\Omega_y \times \Omega_x$. The transform $\mathcal{K}$ can be interpreted as mapping a function $v(x)$ defined on domain $x \in \Omega_x$ to the function $[\mathcal{K}(v)](y)$ defined on domain $y \in \Omega_y$. Recently, it has been proved that the kernel integral operation can be successfully approximated by the attention mechanism of Transformers both theoretically and empirically (Cao, 2021; Kovachki et al., 2021; Guibas et al., 2021; Pathak et al., 2022). Intuitively, let input vectors $X \in \mathbb{R}^{n_x \times d_x}$ and query vectors $Y \in \mathbb{R}^{n_y \times d_y}$, then the attention can be expressed as

$$Attn(Y, X, X) = \sigma(QK^T)V \approx \int_{\Omega_x} (q(Y) \cdot k(x))v(x)dx, \tag{7}$$

where $Q = YW^q \in \mathbb{R}^{n_y \times d_q}$, $K = XW^k \in \mathbb{R}^{n_x \times d_q}$, $V = XW^v \in \mathbb{R}^{n_x \times d_v}$, and $\sigma$ are the query, key, value matries, and softmax function, respectively. Details of the derivation can be found in Appendix A.3. The attention mechanism, the weighted sum of $V$ with the attention matrix $\sigma(QK^T)$, can be interpreted as the kernel integral operation in which the parameterized kernel is approximated by the attention matrix (Cao, 2021; Tsai et al., 2019; Xiong et al., 2021; Choromanski et al., 2021). This attention is also known as cross-attention, where the input vectors are projected to query embedding space by the attention, $Attn(Y, X, X)$. Note that when $X = Y$, the mechanism denotes the self-attention, $Attn(X, X, X)$.

The attention blocks $Attention(Y, X, X)$ used in this paper are described in the Appendix A.4. The term $Attention(Y, X, X)$ used in the following paper stands for the attention blocks. The nonlinear functions $\sigma$, point-wise linear transformations $W_l$, and kernel integral operations $K_l$ in the iterative update (Equation 5) are approximated by feed-forward neural networks, residual connections, and the attention modules, which are conventional modules from Transformer-like architectures (Vaswani et al., 2017). When we consider the attention block as mapping operation $Attention(Y, X, X) : \mathbb{R}^{n_x} \mapsto \mathbb{R}^{n_y}$, it can be used to not only approximate the kernel integral operation but also decouple the cardinality and sampled locations of input vector $X$ and output $Attention(Y, X, X)$.

# 4 APPROACH

## 4.1 PREPROCESSING

**Set representations for the discretizations.** Before starting to construct a model, we reconsider the representations of $a = a|_X \in \mathbb{R}^{n_x \times d_a}$, $u = u|_Y \in \mathbb{R}^{n_y \times d_u}$ and query coordinates $Y_u \in \mathbb{R}^{n_y \times d}$. To compensate the positional information, we follow the common way of existing neural operators literature (Li et al., 2021b; Kovachki et al., 2021), which usually concatenate the position coordinates and the corresponding values as the input representation, $a = \{(x_1, a(x_1)), ..., (x_{n_x}, a(x_{n_x}))\} \in \mathbb{R}^{n_x \times (d + d_a)}$. Then, we treat $a$, $u$, and $Y_u$ as the set representations represented by flattened arrays, without using structured bias (e.g., 2D, 3D-structured arrays, or local neighbor connected graphs) to

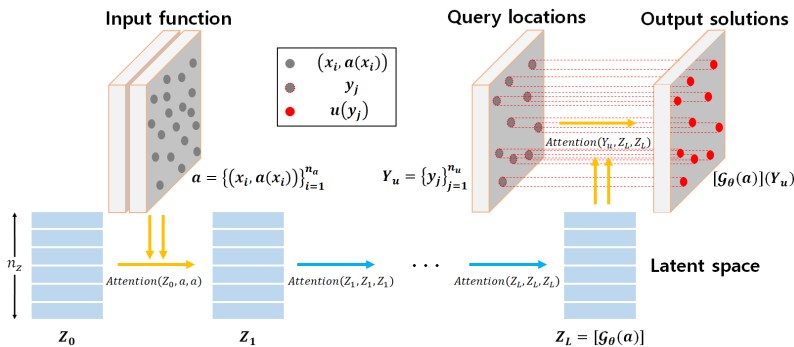

Figure 1: Mesh-independent operator learner.

avoid our model bias toward specific data structure and flexibly process any of the discretization formats of input and output. By the concatenated position coordinates for each value, the output should be permutation-invariant to the representation of the input function $a$, when the set is represented as feature vectors, i.e.,

$$[\mathcal{G}_\theta(\pi a)](Y) = [\mathcal{G}_\theta(a)](Y). \tag{8}$$

Meanwhile, since the $u$ is output of the model at the query coordinates $Y = \{y_1, ..., y_{n_y}\} \in \mathbb{R}^{n_y \times d}$, only the values are used, $u = \{u(y_1), ..., u(y_{n_y})\} \in \mathbb{R}^{n_y \times d_u}$. Since the output $u(y)$ is a function of query coordinates, the permutation operation and the function operations are commutative. Therefore, the output should be permutation-equivariant to the representation of the query coordinates $Y$, when the set is represented as vectors, i.e.,

$$[\mathcal{G}_\theta(a)](\pi Y) = \pi[\mathcal{G}_\theta(a)](Y). \tag{9}$$

Detailed explanations of the permutation-symmetry can be found in Appendix A.3.

**Positional embeddings.** Instead of using raw position coordinates for both inputs and outputs $x_i, y_j \in \mathbb{R}^d$, we concatenate the Fourier embeddings for the position coordinates, which is a common strategy to enrich the positional information (Vaswani et al., 2017; Mildenhall et al., 2020; Tancik et al., 2020; Sitzmann et al., 2020). The positional embeddings exploit sine and cosine functions with frequencies spanning from minimum to maximum frequencies sufficiently covering the Nyquist rates for corresponding dimensions This simple technique provides the model with the capability of representing fine-grained functions or wide-spectral components.

## 4.2 ARCHITECTURE

Following (Lee et al., 2019; Jaegle et al., 2021; 2022; Tang & Ha, 2021), we build our model with a attention-based architecture consisting of encoder-processor-decoder with modality-agnostic encoder and decoder, called mesh-independent operator learner (MIOL) as presented in Figure 1. The significant modifications from the existing neural operators are mostly in the lifting (encoder) and projection (decoder) for detaching the dependence of input and output meshes. The encoder encodes the input function $a$ to fixed smaller number of latent feature vectors (discretization number: $n_x \mapsto n_z$), satisfying permutation-invariance, the processor processes the pair-wise and higher-order interactions between elements of the latent features vectors (discretization number: $n_z \mapsto n_z$), and the decoder decodes the latent features to output solutions at a set of query coordinates $Y_u$ (discretization number: $n_z \mapsto n_y$) satisfying permutation-equivariance. The permutation-symmetric property of attention blocks are described in Appendix A.3.

**Encoder.** We use a cross-attention block as the encoder to encode inputs $a \in \mathbb{R}^{n_x \times (d+d_a)}$ to a smaller fixed number $n_z$ of learnable queries $Z_0 \in \mathbb{R}^{n_z \times d_z}$ (typically $n_z < n_x$), then the result of the block is

$$Z_1 = \mathcal{G}_{enc}(a) = Attention(Z_0, a, a) \in R^{n_z \times d_h}, \tag{10}$$

which is permutation-invariant to the elements of $a$ and independent of the size of the input $n_x$. These properties, which make the model mesh-independent to $a$, are a significant difference from the lifting component in the existing neural operators, which is not permutation-invariant and outputs the same size as the inputs $n_x$ with point-wise transformation. The encoder is also interpreted as

a projection inputs domain discretized by $n_x$ elements to a latent domain consisting of the smaller number of $n_z$ elements, which is called "inducing points" in (Lee et al., 2019) and reduces the computational complexity of the following attentional blocks (Jaegle et al., 2021; 2022).

**Processor.** We use a series of self-attention blocks as the processor each of which takes $z_l \in \mathbb{R}^{n_z \times d_h}$ as the input of the query, key, and value components. Then the output of each self-attention block with $l \in [1, L-1]$ is

$$Z_{l+1} = \mathcal{G}_l(Z_l) = Attention(Z_l, Z_l, Z_l) \in R^{n_z \times d_h}, \tag{11}$$

which is permutation-equivariant to the elements of $Z_l$, therefore the permutation-invariant property to the elements of $a$ is preserved through successive modules. Also, the results $Z_{l+1}$ have fixed discretization format with fixed ordering and number of elements $n_z$ which is decoupled from discretization format of the input function $n_x$ and output functions $n_y$. Due to the decoupling property, the whole architecture can not only capture the global interactions by the processor but is also applicable to mesh-independent operator learning independent of discretization formats of the input and output functions.

**Decoder.** We use a cross-attention block as the decoder to decode the latent vectors from the processor $Z_L \in \mathbb{R}^{n_z \times d_h}$ at query coordinates $Y \in \mathbb{R}^{n_y \times d}$. Then the final output of the entire architecture is

$$[\mathcal{G}_\theta(a)](Y) = \mathcal{G}_{dec}(Z_L) = Attention(Y, Z_L, Z_L) \in \mathbb{R}^{n_y \times d_u}, \tag{12}$$

which is permutation-equivariant to the elements of $Y_u$, therefore every solution $u_j$ corresponds to query coordinates $y_j$. This property makes the model applicable to any arbitrary discretization format of $Y$. Since the result from the processor is independent of the discretization format of input function $a$, the model is also applicable to any arbitrary discretization format of $a$.

## 5 EXPERIMENTS

### 5.1 EXPERIMENTAL SETTINGS

**Datasets.** Our experiments are conducted on several PDE benchmarks datasets following (Li et al., 2021b), such as Burgers equations, Darcy flow, and Navier-Stokes equations, and 3D spherical shallow water dataset following (Yin et al., 2022), to investigate the flexibility and generalization abilities of MIOL through various downstream tasks. Details of the equations and the problems are described in Appendix A.5.

**Baselines.** We took several representative architectures as baselines for PDE benchmarks problems, such as the original graph neural operator (GNO) (Li et al., 2020a), the multipole graph neural operator (MGNO) (Li et al., 2020b), a neural operator based on low-rank decomposition (LNO) (Li et al., 2021b), the deep operator network (DeepONet) (Lu et al., 2019), and the Fourier neural operator (FNO) (Li et al., 2021b), and baselines for a 3D spherical shallow water problem, such as message passing PDE solver (MP-PDE) (Brandstetter et al., 2022), and dynamic-aware implicit neural representation (DINO) (Yin et al., 2022).

While we follow the original training procedures of (Li et al., 2021b), we take extended tasks with test sets varying discretization points for input $a$ and solution $u$ (or query coordinates $y$) to evaluate the generalization abilities of architectures for input and output formats. Results for experiments already discussed on these baselines were obtained from the related literature, and results for the extended tasks that have not been discussed before have been reproduced from their original codes. Note that the 'n/a' result will occur when the baselines cannot be applicable for some downstream tasks. The implementation details and additional experimental results can be found in the Appendix A.6, A.7, respectively.

### 5.2 RESULTS

**Original benchmark tasks.** Original benchmarks tasks on Burgers' equation and Darcy flow for different resolutions are presented in Figure 2, and the corresponding qualitative results are presented in Table 6, and Table 7 (Appendix A.7), where all of the results except ours are brought from (Li et al., 2021b). The results show that our MIOL outperforms almost all neural operators at

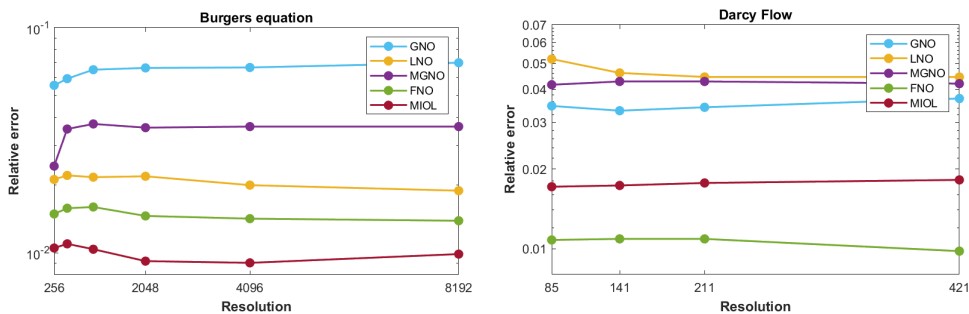

Figure 2: Benchmarks on (a) Burgers' equation, and (b) Darcy Flow for different resolutions.

every resolution in a mesh-invariant way, except FNO in case of Darcy flow. This is because FNO explicitly exploits structural bias of the 2D structured array with equally spaced grids, while MIOL does not use the structural bias. For example, in the case of 2D Darcy flow, when the input values $a(x_1, x_2)$ are given at $85 \times 85$ grid points (where the dimension of position coordinates $d = 2$ and the corresponding function values $d_a = 1$), MIOL flattens to treat the batch-wise inputs as flatten arrays like $a \in R^{b \times 7225 \times 3}$, while FNO treats them as 2D structured-arrays like $a \in R^{b \times 85 \times 85 \times 3}$.

**Tasks for evaluating mesh-independence.** To distinguish the specific generalization abilities of the existing architectures and to show that our proposed MIOL has all the generalization abilities and is robustly applicable even for tasks that the existing architectures are unable to solve, we compare MIOL with two main baselines, DeepONet (Lu et al., 2019) and FNO (Li et al., 2021b), by the following selective tasks presented in Table 6.

- **Task 1: Supervised learning.** This is the original setting of the existing operator learning models which evaluate the performance under the supervised learning settings. The results are included in the original benchmark tasks, which shows that the MIOL is competitive with others and slightly worse than FNO in case of Darcy flow.

- **Task 2: Mesh-independence to input, but requiring the same grids for $a = u$.** This task, discussed in previous works of neural operators (Li et al., 2021b), evaluates the performance under zero-shot learning settings where the solutions are provided in unseen discretization during training. However, requiring the same grids for input and output functions, the solutions $u$ of neural operators are only able to be queried at the same grids of $a$. DeepONets are not commonly applicable due to requirements of fixed input mesh, while FNO and MIOL are robustly applicable and performances are almost mesh-invariant.

- **Task 3: Mesh-independence to output, but requiring the same grids for train $a = $ test $a$.** This task, discussed in previous works on operator networks (Lu et al., 2019), evaluates the performance under zero-shot learning settings where the solutions are provided in unseen discretization during training. However, operator networks are only applicable when the discretization formats (same numbers and locations) of the input function remain the same during training and testing. FNO are not commonly applicable due to the requirements of the same grids for $a = u$, whereas DeepONet and MIOL are robustly applicable, and their performances are almost mesh-invariant.

- **Task 4: Randomly masked input, and queried at all points.** This task, which has not been discussed before, evaluates the performance under a zero-shot learning setting, wherein the discretizations of input and output functions are allowed to have different numbers and be in random locations. The settings of the random mask for every input function and queried at all given points are sufficient to evaluate whether the solution of the model is independent of the discretization of the input function and can be queried at any point. DeepONet and FNO are not applicable because of their restrictions, whereas MIOL is robustly applicable.

**Predictions at any query point from the varying number and irregular distributed inputs.** Additionally, Figure 3 visualizes the predictions of MIOL at all given query coordinates from the varying number and irregularly distributed inputs in Burgers' equation (left), and Darcy Flow (right). This visualization is reminiscent of few-shot regression problems (Finn et al., 2017; Kim et al., 2019) for $u(y)$ at query locations $y$, where the support set is given by relatively few samples of discretized inputs $a$ that should be mapped to the output space $u$ through a series of integral transforms. As presented, the more shots are given, the closer the predictions are to the ground truth.

Table 2: Relative $L^2$ errors on Burgers' equation and Darcy flow under different settings.

| Task | Train grids | Test grids | | Models | | |
|------|-------------|------------|------|----------|-----|-------------|
| | $a, u$ | $a$ | $u$ | DeepONet | FNO | MIOL (ours) |
| | | Burgers' equation | | | | |
| (1) | | 1024 | 1024 | 0.1582 | 0.0160 | **0.0104** |
| (2) | 1024 | 8192 | 8192 | n/a | 0.0139 | **0.0090** |
| (3) | | 1024 | 8192 | 0.1584 | n/a | **0.0106** |
| (4) | | 512 (50% mask) | 1024 | n/a | n/a | **0.0479** |
| | | Darcy flow | | | | |
| (1) | | 85×85 | 85×85 | 0.0765 | **0.0108** | 0.0172 |
| (2) | 85×85 | 421×421 | 421×421 | n/a | **0.0098** | 0.0170 |
| (3) | | 85×85 | 421×421 | 0.0766 | n/a | **0.0173** |
| (4) | | 3612 (50% mask) | 85×85 | n/a | n/a | **0.0272** |

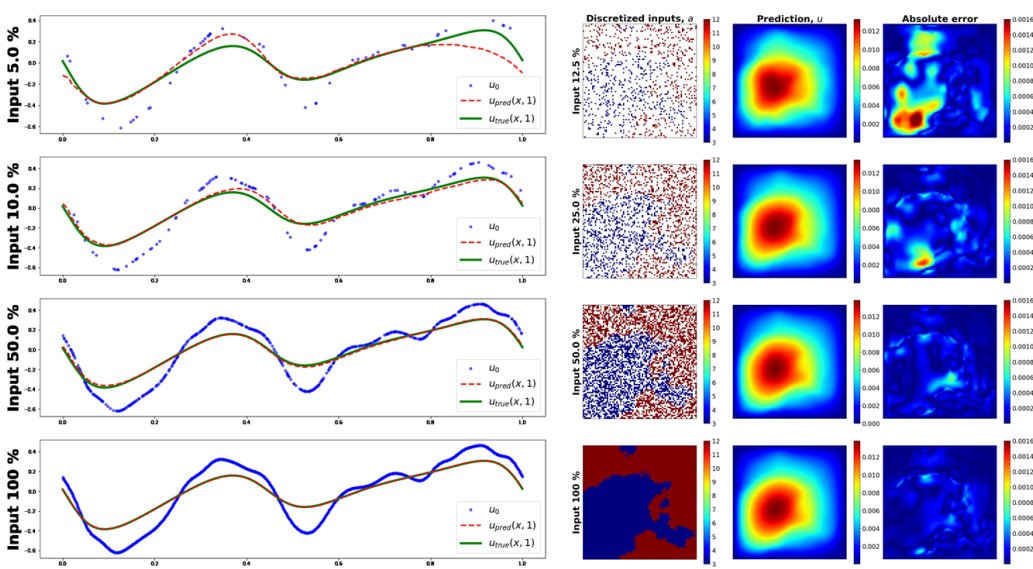

Figure 3: Predictions of MIOL at all query coordinates from the varying number and irregular distributed inputs on Burgers' equation (left), and Darcy Flow (right).

**Applications to time-stepping systems.** For the time-dependent PDEs, we assumed that the systems follow a Markov process in which the state at time $t + dt$ is described as $w_{t+dt} = \mathcal{G}_{dt} \cdot w_t$, where the operator $\mathcal{G}_{dt}$ during $dt$ is not dependent on time (Li et al., 2021a). When we set the $dt = 1$, the state of the system at time $t$ is denoted as $w_t = \mathcal{G}_{dec} \cdot [\mathcal{G}_{t-1} \cdot \cdots \cdot \mathcal{G}_1] \cdot \mathcal{G}_{enc}(w_0)$ where $\mathcal{G}_{enc}$ and $\mathcal{G}_{dec}$ are the encoder and decoder implemented by the cross-attention blocks, respectively, and $\mathcal{G}_{dt} = \mathcal{G}_1 = \cdots = \mathcal{G}_{t-1}$ are identical processors implemented by the same self-attention blocks. The goal is to learn the mapping from the initial state up to time $T$, $\mathcal{G} : w_0|_X \mapsto w|_{X \times (0,T]}$. The models were trained by minimizing the object $E_{w_0 \sim \mu} \left[ \frac{1}{T} \sum_{t=0}^{T-1} \mathcal{L}(\mathcal{G}_{dec} \cdot \mathcal{G}_{dt} \cdot \mathcal{G}_{enc}(w_t), w_{t+1}) \right]$. Figure 4 visualizes that a initial states were encoded and processed iteratively with the same processors in latent space. The latent features were decoded to output the predictions of the state fields at the corresponding time. The quantitative results for Navier-Stokes equation and spherical shallow water are presented in Table 8, and Table 9, respectively.

**Performances according to input masking ratios.** Relative $L^2$ errors according to input masking ratios (%) on the extended tasks are presented in Figure 5 and Table 3. The discretized inputs are randomly masked with corresponding ratios. The errors are consistently low when the masking ratio is lower than approximately 50%, where the performances according to input masking ratios are found in Figure 5 and Table 3. The efficient set encoder can aggregate sufficient information from the subsampled input $a_i^{(p)}$ to make sufficiently small discretization errors from Equation 4.

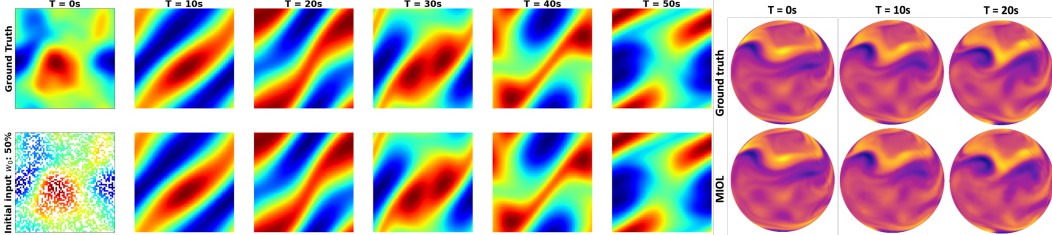

Figure 4: Initial states were encoded, processed, and decoded to output the predictions of states for the Navier-Stokes equation (left) and 3D spherical shallow water (right) at the corresponding time. Since MIOL can process arbitrary discretization formats of input and output, it can be applicable even when the initial input states are randomly masked with masking ratio $p = 50\%$ (left) or can be queried even at finer grids on the 3D spherical surface (right).

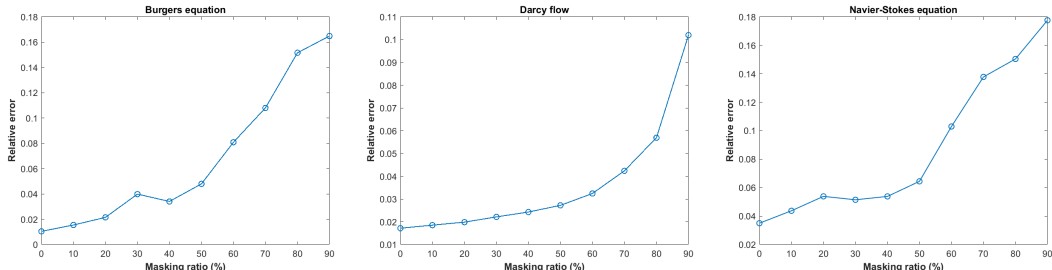

Figure 5: Relative $L^2$ errors according to input masking ratios (%) on (a) Burgers' equation, (b) Darcy Flow, and (c) Navier-Stokes equations.

Table 3: Relative $L^2$ errors according to input masking ratios (%).

| Masking ratio | 0% | 10% | 20% | 30% | 40% | 50% | 60% | 70% | 80% | 90% |
|---|---|---|---|---|---|---|---|---|---|---|
| Burgers | 0.0104 | 0.0154 | 0.0214 | 0.0399 | 0.0340 | 0.0479 | 0.0809 | 0.1081 | 0.1516 | 0.1649 |
| Darcy flow | 0.0172 | 0.0185 | 0.0198 | 0.0221 | 0.0243 | 0.0272 | 0.0324 | 0.0424 | 0.0569 | 0.1019 |
| Navier-Stokes | 0.0349 | 0.0437 | 0.0538 | 0.0514 | 0.0538 | 0.0644 | 0.1030 | 0.1378 | 0.1504 | 0.1776 |

## 6    CONCLUSIONS AND LIMITATIONS

**Conclusions.** In this study, we raise potential issues with existing operator learning models for PDEs from the perspective of discretization for the continuous input/output functions of the systems when the observations are irregular and have discrepancies between training and testing measurement formats. To solve these issues, we constructed a fully-attentional architecture called MIOL, which treats discretized functions as set-valued data without prior data structures and structurally separates dependencies on input and output meshes. MIOL is evaluated on the original tasks and extended downstream tasks and compared with other existing representative models. The results show that our model is not only competitive in original operator learning tasks but also robustly applicable in extended tasks which are natural consequences of continuous solutions for physical systems but are not compatible with existing representative models.

**Limitations.** First, although our main contribution is to construct an operator learning model to be flexible in measurement formats, the higher the dimension in which the measurement points can be placed, the performances of our model tested on some original benchmarks are slightly worse than the problem-specific model (details in Appendix A.7), and the computational costs to obtain flexibility grow exponentially (in Table 5). It could be an important direction for future work to address the efficiency and scaling issues of discretization numbers of the input and output functions. Second, the experimental setting was far from a real-world problem. Applications to realistic scenarios are an interesting direction for future research. For example, we might consider using a model trained on the Navier-Stokes dataset to analyze the real fluid flow along complex geometries regarded as the same phenomena with different measurement formats.

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

# A  APPENDIX

## A.1  ARCHITECTURE SCHEMATICS

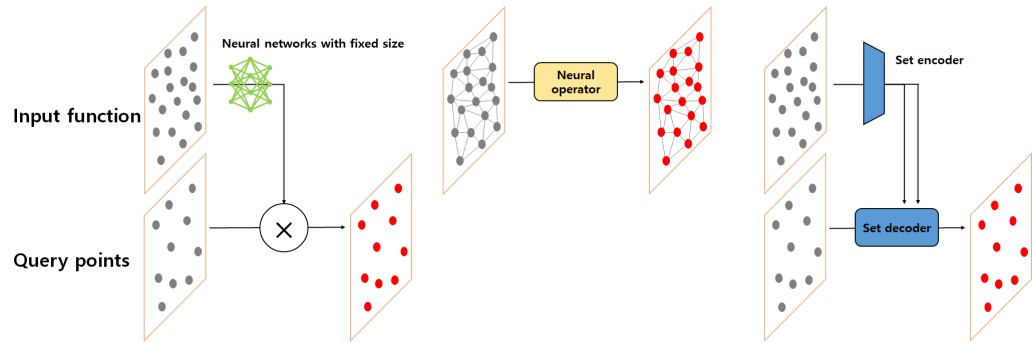

Figure 6: Schematics of operator learning architectures, such as operator networks (left), neural operators (middle), and our proposed MIOL (right). For each respective formulation, the gray circles are given and the red circles are the corresponding outputs.

## A.2  RELATED WORKS

**Neural networks for PDEs.** There has been increasing interest in utilizing neural networks for PDEs that can be divided into several lines. In (Raissi et al., 2019; Sirignano & Spiliopoulos, 2018; Karniadakis et al., 2021), neural networks were used as approximations of the solution when given the boundary conditions and collocations points constrained to known governing equations of the system, called physics-informed neural networks. It can be an alternative to traditional PDE solvers, yielding mesh-independent and relatively high-fidelity solutions. However, these methods require full knowledge of the PDEs, and the trained model is usually not reusable for new systems, otherwise requires expensive re-training for every new boundary condition. Meanwhile, several studies have been conducted to learn reusable governing operators for generalization to new systems. A universal operator approximation theorem (Chen & Chen, 1995) was originally implemented by single-layer neural networks, and (Lu et al., 2019; 2021) develop the theorem by extending the architectures with current deep networks, called DeepONets, leading them to be more expressive. Neural operators have been originally implemented by message passing on graphs (Li et al., 2020a;b). However, the graph-based neural operator was limited due to its stability issues and the quadratically growing costs for computing the integral operator (You et al., 2022; Li et al., 2021b). Fourier neural operators (FNO), which utilize fast Fourier transform (FFT) to efficiently compute the integral operators in Fourier space, have been widely used as successful models for operator learning in a range of science and engineering for their efficacy and accuracy (Li et al., 2021b; Yang et al., 2021; Pathak et al., 2022). Furthermore, (Gupta et al., 2021) uses multi-wavelet transform to approximate the kernel of the operators. Close to our works, (Cao, 2021) use Transformer-style architectures to approximate the integral operation, but they use problem-specific feature extractors and decoding regressors which make the model biased toward specific equations.

**Set representation learning.** In recent years, it has been witnessed that neural networks on sets gained attention starting from learning simple operations on unordered sets (e.g., sum or max) and processing the point clouds (Zaheer et al., 2017; Ravanbakhsh et al., 2016; Qi et al., 2017a;b). The elements of the set are transformed into new representations through point-wise feed-forward neural networks which are permutation-equivariance. Then, the representation of the set is obtained by aggregation across the entire elements of the set by permutation-invariant functions, such as sum-pooling (Zaheer et al., 2017) or max-pooling (Qi et al., 2017a;b). However, a theoretical study has been investigated to claim the limitations of the representation power of the methods (Wagstaff et al., 2019; Jurewicz & Derczynski, 2021). Additionally, the methods make it difficult for the model to learn pair-wise and higher-order interactions between the elements of the sets, which are lost during those pooling operations. More recently, several groups have suggested more sophisticated models for treating set-valued inputs through modifying Transformers (Vaswani et al., 2017). Set Transformer uses a cross-attention module considered as learnable parameterized pooling operations to project set-valued input into smaller latent vectors, which ensure handling arbitrary cardinality and permutation of the inputs (Lee et al., 2019). Perceiver uses a similar strategy by emphasizing

the efficiency of computational cost and the property of modality-agnostic which can be applied to a wide range of large-scale input modalities, such as images, point clouds, audio, and video (Jaegle et al., 2021). Furthermore, beyond simple outputs like classification, Perceiver IO uses another cross-attention module to decode complex outputs modalities which can be applied to the domains of language, optical flow, audio-visual sequence, and game environments (Jaegle et al., 2022). A similar strategy is also applied to reinforcement learning (Tang & Ha, 2021).

### A.3 PERMUTATION-SYMMETRY AND ATTENTION

**Detail explanation of equation 7.** Here is a brief explanation of the approximation with the integral for the cross-attention mechanism, where the softmax for the attention matrix is ignored for simplicity.

$$
Attn(Y, X, X) = \begin{bmatrix} Attn(y_1, X, X) \\ \vdots \\ Attn(y_{n_y}, X, X) \end{bmatrix} = \begin{bmatrix} q(y_1) \\ \vdots \\ q(y_{n_y}) \end{bmatrix} \begin{bmatrix} k_1(x_1) & \dots & k_1(x_{n_x}) \\ \vdots & \ddots & \vdots \\ k_{d_q}(x_1) & \dots & k_{d_q}(x_{n_x}) \end{bmatrix} \begin{bmatrix} v(x_1) \\ \vdots \\ v(x_{n_x}) \end{bmatrix}
$$

$$
= \begin{bmatrix} q(y_1) \cdot k(x_1) & \dots & q(y_1) \cdot k(x_{n_x}) \\ \vdots & \ddots & \vdots \\ q(y_{n_y}) \cdot k(x_1) & \dots & q(y_{n_y}) \cdot k(x_{n_x}) \end{bmatrix} \begin{bmatrix} v(x_1) \\ \vdots \\ v(x_{n_x}) \end{bmatrix} = \begin{bmatrix} \sum_{i=1}^{n_x} (q(y_1) \cdot k(x_i)) v(x_i) \\ \vdots \\ \sum_{i=1}^{n_x} (q(y_{n_y}) \cdot k(x_i)) v(x_i) \end{bmatrix}
$$

$$
\approx \begin{bmatrix} \int_{\Omega_x} (q(y_1) \cdot k(x)) v(x) dx \\ \vdots \\ \int_{\Omega_x} (q(y_{n_y}) \cdot k(x)) v(x) dx \end{bmatrix} = \int_{\Omega_x} (q(Y) \cdot k(x)) v(x) dx.
$$

(13)

Here, the discretization of input is $\{x_1, ..., x_{n_x}\}$ (as key and value vectors), and it can be changed to the discretization of output $\{y_1, ..., y_{n_y}\}$ (as query vectors) with cardinality changed from $n_x$ to $n_y$. Using this mechanism, we can detach the dependences of discretization formats of input and output from the processor, by encoding arbitrary discretization $\{x_1, ..., x_{n_x}\}$ to a fixed size $(n_z)$ of learnable latent set vectors, and decoding the latent set vectors to output arbitrary discretization $\{y_1, ..., y_{n_y}\}$. The discretization number is varied as $n_x$ (arbitrary) $\rightarrow n_z$ (fixed) $\rightarrow n_y$ (arbitrary).

**Permutation invariance and equivariance.** For a function whose input can be represented by a set, there can be two interesting symmetries with respect to any permutations; permutation-invariance, and permutation-equivariance. Let $F$ is the function with set-valued inputs $X = \{x_1, x_2, ..., x_n\}$, $x_i \in \mathbb{R}^d$. Since deep learning models can not directly treat the set-valued input, the input data is provided as ordered vectors representing one of $n!$ permutations of $X$. In practice, with arbitrary permutation action $\pi$ to the first dimension of the matrix-represented $X \in \mathbb{R}^{n \times d}$, the permutation-invariance and equivariance are defined as,

$$
\begin{aligned}
F(\pi X) &= F(X), &\text{(permutation-invariance)}, \\
F(\pi X) &= \pi F(X), &\text{(permutation-equivariance)},
\end{aligned}
$$

(14)

where the output of the permutation-invariant function does not depend on ordering and the cardinality of the input set, while the ordering and the cardinality of the output of the permutation-equivariant function remain consistent with those of the input set. A full mathematical definition and details of the properties can be found in (Zaheer et al., 2017; Murphy et al., 2019; Wagstaff et al., 2021; Maron et al., 2020).

**Permutation-symmetric property of attention.** Here, we attempt to explain that the attention module $Attn(Y, X, X)$ in Equation 7 is permutation-invariant to $X$ and permutation-equivariant to $Y$. Let the permutation matrix to the element (first) dimension, $P \in \mathbb{R}^{n \times n}$. First, when the

permutation matrix $P$ is multiplied to the $X$, the output of the attention is

$$
\begin{aligned}
Attn(Y, PX, PX) &= \sigma(QK^T)V \\
&= \sigma\left(YW^q \left(PXW^k\right)^T\right) PXW^v \\
&= \sigma\left(YW^q \left(XW^k\right)^T P^T\right) PXW^v \\
&= \sigma\left(YW^q \left(XW^k\right)^T\right) \left(P^T P\right) XW^v \\
&= \sigma\left(YW^q \left(XW^k\right)^T\right) XW^v \\
&= Attn(Y, X, X),
\end{aligned}
\tag{15}
$$

which states that the $Attn(Y, X, X)$ is permutation-invariant to $X$. Also, when the permutation matrix $P$ is multiplied to the $Y$, the output of the attention is

$$
\begin{aligned}
Attn(PY, X, X) &= \sigma(QK^T)V \\
&= \sigma\left(PYW^q \left(XW^k\right)^T\right) XW^v \\
&= P\sigma\left(YW^q \left(XW^k\right)^T\right) XW^v \\
&= P Attn(Y, X, X),
\end{aligned}
\tag{16}
$$

which states that the $Attn(Y, X, X)$ is permutation-equivariant to $Y$. Thus, it is easily proved that self-attention $Attn(X, X, X)$ is permutation-equivariant to $X$,

$$
Attn(PX, PX, PX) = P Attn(X, PX, PX) = P Attn(X, X, X).
\tag{17}
$$

## A.4 ATTENTION BLOCKS

Following (Lee et al., 2019; Jaegle et al., 2021; 2022), mesh-independent operator learner (MIOL) consists of two types of attention blocks, cross- and self-attention blocks, which implement the respective attention mechanisms. The attention blocks have the following shared structure, which takes two input arrays, a query input $Y \in \mathbb{R}^{n_y \times d_y}$ and a key-value input $X \in \mathbb{R}^{n_x \times d_x}$,

$$
\begin{aligned}
O &= Y + Attn(LayerNorm(Y), LayerNorm(X), LayerNorm(X)), \\
Attention(Y, X, X) &= O + FF(LayerNorm(O)),
\end{aligned}
\tag{18}
$$

where $LayerNorm$ is layer normalization (Ba et al., 2016), $FF$ consists of two point-wise feed-forward neural networks with a GELU nonlinearity (Hendrycks & Gimpel, 2016), and the exact calculation of attention $Attn$ is

$$
Attn(X^q, X^k, X^v) = softmax\left(\frac{QK^T}{\sqrt{d_q}}\right) V,
\tag{19}
$$

where $Q = X^q W^q \in \mathbb{R}^{n_y \times d_q}$, $K = X^k W^k \in \mathbb{R}^{n_x \times d_q}$, and $V = X^v W^v \in \mathbb{R}^{n_x \times d_v}$ for a single headed attention. In the case of multi-headed attention, several outputs from different learnable parameters are concatenated and projected with the linear transformation. Here, the output of cross-attention block $Attention(Y, X, X)$ is also permutation-invariant to $X$ and permutation-equivariant to $Y$, and the output of self-attention block $Attention(X, X, X)$ is also permutation-equivariant to $X$, because the other modules operate in a point-wise way, the properties of symmetry for the attention modules are preserved for the corresponding attention blocks.

## A.5 BENCHMARKS

**Burgers' equation.** First, we consider a benchmark problem of 1D Burgers' equation which is a non-linear parabolic PDE combining the terms of convection and diffusion. The equation with periodic boundary conditions is

$$
\begin{aligned}
\partial_t u(x, t) + \partial_x (u^2(x, t)/2) &= \nu \partial_{xx} u(x, t), & x \in (0, 1), t \in (0, 1] \\
u(x, 0) &= u_0(x), & x \in (0, 1)
\end{aligned}
$$

where $u_0 \sim \mu$ is the initial state generated from $\mu = \mathcal{N}(0, 625(-\Delta + 25I)^{-2})$ and $\nu = 0.1$ is the viscosity coefficient. The goal of operator learning is to learn mapping the initial state to the solution at time one, $\mathcal{G} : u_0 \mapsto u(\cdot, 1)$.

**Darcy flow.** Second, we consider another benchmark problem of 2D steady-state Darcy flow which is a second-order elliptic PDE describing the flow of fluid through a porous medium. The equation of Darcy flow on the unit box is

$$-\nabla \cdot (a(x)\nabla u(x)) = f(x), \qquad x \in (0,1)^2$$
$$u(x) = 0, \qquad x \in \partial(0,1)^2$$

where $u$ is density of the fluid, $a \sim \mu$ is the diffusion field generated from $\mu = \mathcal{N}(0, (-\Delta + 9I)^{-2})$ with fixed forcing function $f = 1$. The goal of operator learning is to learn mapping the diffusion field to the solution of the density, $\mathcal{G} : a \mapsto u$.

**Navier-Stokes equation.** Third, we consider another benchmark problem of 2D Navier-Stokes equation describing the dynamics of a viscous, incompressible fluid. The equation in vorticity form on the unit torus is

$$\partial_t w(x,t) + u(x,t) \cdot \nabla w(x,t) = \nu \Delta w(x,t) + f(x), \qquad x \in (0,l)^2, t \in (0,T]$$
$$\nabla \cdot u(x,t) = 0, \qquad x \in (0,l)^2, t \in [0,T]$$
$$w(x,0) = w_0(x), \qquad x \in (0,l)^2$$

where $u$ is the velocity field, $w = \nabla \times u$ is the vorticity field, $w_0 \sim \mu$ is the initial vorticity field generated from $\mu = \mathcal{N}(0, 7^{3/2}(-\Delta + 49I)^{-2.5})$ with periodic boundary conditions, $\nu$ is the viscosity coefficient and forcing function is kept $f(x) = 0.1 (\sin(2\pi(x_1 + x_2)) + \cos(2\pi(x_1 + x_2)))$.

## A.6 IMPLEMENTATION DETAILS

Table 4: Implementation details for training.

| System | Burgers' equation | Darcy flow | Navier-Stokes equation |
|---|---|---|---|
| Encoder | | | |
| Input function values, $a\|_{X_a} \in \mathbb{R}^{n_x \times d_a}$ | 1024×1 | 7225×1 | 4096×1 |
| Positional embedding | 1024×129 | 7225×130 | 4096×50 |
| Frequency bins, $K$ | 64 | 32 | 12 |
| Number of heads | 4 | 1 | 1 |
| Inputs, $a \in \mathbb{R}^{(d+d_a)}$ | 1024×130 | 7225×131 | 4096×51 |
| Processor | | | |
| Learnable queries, $Z_0 \in \mathbb{R}^{n_z \times d_z}$ | 256×64 | 256×64 | 128×64 |
| Latent channels, $d_h$ | 64 | 64 | 64 |
| Number of heads | 8 | 8 | 8 |
| Number of blocks, $L$ | 1 | 4 | 2 |
| Decoder | | | |
| Query coordinates, $Y_u \in \mathbb{R}^{n_y \times d}$ | 1024×1 | 7225×2 | 4096×2 |
| Positional embedding | 1024×129 | 7225×130 | 4096×50 |
| Outputs, $u \in \mathbb{R}^{n_y \times d_u}$ | 1024×1 | 7225×1 | 4096×1 |

**Burgers' equation.** The positional embeddings at position coordinate $x \in [0,1]$ from both input/output domain consists of $[x, \sin(f_k \pi x), \cos(f_k \pi x)]$, where $f_k$ are from equally spaced 64 frequencies from min 1 to max resolution 64 are used, resulting to have $129 = 1 + 2 \times 64$ dimensions. For the encoder, a 4-headed cross-attention module is used, and the number of the elements (first) and channel (second) dimension of latent space are 256 and 64, respectively. For the processor, a 8-headed self-attention module with the same channel dimension of latent's is used. For the decoder, a 4-headed cross-attention module with the channel dimension of 64 is used.

**Darcy flow.** The positional embeddings at position coordinate $(x_1, x_2) \in [0,1]^2$ of diffusion field $a(x_1, x_2)$ consists of $[x_1, x_2, \sin(f_k \pi x_1), \sin(f_k \pi x_2), \cos(f_k \pi x_1), \cos(f_k \pi x_2)]$, where $f_k$ are from equally spaced 32 frequencies from min 1 to max resolution 32 are used, resulting to have

$130 = 2 \times (1 + 2 \times 32)$ dimensions. For the encoder, a single-headed cross-attention module is used, and the number of the elements (first) and channel (second) dimension of latent space are 256 and 64, respectively. For the processor, four 8-headed self-attention modules with the same channel dimension of latent's are used. For the decoder, a single-headed cross-attention module with the channel dimension of 64 is used.

**Navier-Stokes equation.** The positional embeddings at position coordinate $(x_1, x_2) \in [0, 1]^2$ of the vorticity field $w(x_1, x_2)$ consists of $[x_1, x_2, \sin(f_k \pi x_1), \sin(f_k \pi x_2), \cos(f_k \pi x_1), \cos(f_k \pi x_2)]$, where $f_k$ are from equally spaced 32 frequencies from min 1 to max resolution 32 are used, resulting to have $50 = 2 \times (1 + 2 \times 12)$ dimensions. For the encoder, a single-headed cross-attention module is used, and the number of the elements (first) and channel (second) dimension of latent space are 128 and 64, respectively. For the processor, two 8-headed self-attention modules with the same channel dimension of latent's are used. For the decoder, a single-headed cross-attention module with the channel dimension of 64 is used.

**Spherical shallow water.** The position coordinate $(x_1, x_2, x_3)$ are transformed from spherical coordinates of latitude and longitude $\phi, \theta \in [-\pi/2, \pi/2] \times [-\pi, \pi]$. Therefore, the positional embeddings can be $[x_1, x_2, x_3, \sin(f_k \pi x_1), \sin(f_k \pi x_2), \sin(f_k \pi x_3), \cos(f_k \pi x_1), \cos(f_k \pi x_2), \cos(f_k \pi x_3)]$, where $f_k$ are from equally spaced 24 frequencies from min 1 to max resolution 32 are used, resulting to have $147 = 3 \times (1 + 2 \times 24)$ dimensions. For the encoder, a single-headed cross-attention module is used, and the number of the elements (first) and channel (second) dimension of latent space are 256 and 128, respectively. For the processor, two 8-headed self-attention modules with the same channel dimension of latent's are used. For the decoder, a single-headed cross-attention module with the channel dimension of 128 is used.

**Optimization.** For all systems, the experiments are conducted on a GTX TITAN X GPU and we use Adam optimizer (Kingma & Ba, 2014) to train MIOL for 500 epochs with initial learning rate of 0.001 which is halved every 100 epochs. Batch sizes for Burgers' equation and Navier-Stokes equations are set to 20, and for Darcy flow is set to 10.

**Computational complexity comparisons.** In Table 5, we compare MIOL with the baselines in perspective of the training time, memory usage, and number of parameters on the benchmarks. The training time is measured during one epoch of each benchmarks training set, the memory usage indicates the GPU memory usage for the active Python process recorded from nvidia-smi command. The time and memory cost of MIOL is on par with both baselines in the case of Burgers' equation but increases drastically compared with both baselines in the case of Darcy flow. By flattening the discretized inputs into a 1D sequence (identical point set) before processing the inputs and query coordinates, which removes any intrinsic locality of data, MIOL is the modality-agnostic architecture that does not exploit the structural bias of data (such as a 2D structured array). Therefore, for instance, MIOL regards the inputs of Darcy flow as 7,225 sizes of 1D sequence instead of $85 \times 85$ sizes of 2D structured data, which leads to exponential growth computational cost compared to encoding 1D data such as inputs of Burgers' equations. Exponentially more resources are required for MIOL to have the capability to handle the more highly structured data in a modality-agnostic way for flexible input formats and query coordinates. Meanwhile, while attention-based architecture consists of QKV modules where the parameters are shared across the input elements, kernel-based architecture needs a large number of parameters when the dimension of data increases due to the dimension of the kernel increasing along the dimension of data. Therefore, although the number of parameters of FNO is smaller than MIOL in the case of Burgers' equations, it becomes much larger than those of MIOL. In summary, MIOL has fewer parameters but becomes more expressive when paying a large amount of computational cost for gaining the flexibility to data formats, while FNO has a huge number of parameters, which already include some prior data structure, requiring less training cost but could be overfitted to the data structure.

## A.7 ADDITIONAL RESULTS

**Quantitative results for 1D Burgers' equation, 2D Darcy flow, Navier-Stokes equation, and 3D spherical shallow water.**

**Comparisons with other set-encoder.** The comparison model consists of encoder-processor-decoder like MIOL, where encoder is replaced by Deepsets (Zaheer et al., 2017) variant with average-pooling operations, while processor and decoder are the same as MIOL, called Deepsets-

Table 5: Computational complexity comparisons with baselines.

| Model | Training time per epoch | Memory usage | Number of parameters |
|---|---|---|---|
| Burgers' equation | | | |
| DeepONet | 0.645 s | 1.34 GB | 107,405 |
| FNO | 0.871 s | 1.45 GB | 57,825 |
| MIOL | 0.715 s | 1.63 GB | 125,204 |
| Darcy flow | | | |
| DeepONet | 0.732 s | 1.40 GB | 417,550 |
| FNO | 1.881 s | 1.96 GB | 1,188,353 |
| MIOL | 8.843 s | 5.68 GB | 401,514 |

Table 6: Benchmarks on 1D Burgers' equation

| Models | $s = 256$ | $s = 512$ | $s = 1024$ | $s = 2048$ | $s = 4096$ | $s = 8192$ |
|---|---|---|---|---|---|---|
| FCN | 0.0958 | 0.1407 | 0.1877 | 0.2313 | 0.2855 | 0.3238 |
| PCANN | 0.0398 | 0.0395 | 0.0391 | 0.0383 | 0.0392 | 0.0393 |
| GNO | 0.0555 | 0.0594 | 0.0651 | 0.0663 | 0.0666 | 0.0699 |
| LNO | 0.0212 | 0.0221 | 0.0217 | 0.0219 | 0.0200 | 0.0189 |
| MGNO | 0.0243 | 0.0355 | 0.0374 | 0.0360 | 0.0364 | 0.0364 |
| FNO | 0.0149 | 0.0158 | 0.0160 | 0.0146 | 0.0142 | 0.0139 |
| MIOL (ours) | 0.0105 | 0.0109 | 0.0104 | 0.0092 | 0.0090 | 0.0099 |

Table 7: Benchmarks on 2D Darcy Flow

| Models | $s = 85$ | $s = 141$ | $s = 211$ | $s = 421$ |
|---|---|---|---|---|
| FCN | 0.0253 | 0.0493 | 0.0727 | 0.1097 |
| PCANN | 0.0299 | 0.0298 | 0.0298 | 0.0299 |
| GNO | 0.0346 | 0.0332 | 0.0342 | 0.0369 |
| LNO | 0.0520 | 0.0461 | 0.0445 | - |
| MGNO | 0.0416 | 0.0428 | 0.0428 | 0.0420 |
| FNO | 0.0108 | 0.0109 | 0.0109 | 0.0098 |
| MIOL (ours) | 0.0172 | 0.0173 | 0.0177 | 0.0182 |

Table 8: Relative $L^2$ errors on Navier-Stokes equation under different settings.

| Train grids | Test grids | | Models | |
|---|---|---|---|---|
| $a, u$ | $a$ | $u$ | FNO | MIOL (ours) |
| 64×64 | 64×64 | same with $a$ | **0.0110** | 0.0349 |
| | 2048 (50% mask) | 64×64 | n/a | **0.0644** |

Table 9: $L^2$ super-resolution test errors on spherical shallow water

| Model | $0 < T \leq 10$ | $10 < T \leq 20$ |
|---|---|---|
| I-MP-PDE (Brandstetter et al., 2022) | 1.908e-3 | 7.240e-3 |
| DINO (Yin et al., 2022) | 1.063e-4 | 6.466e-4 |
| MIOL (ours) | 4.524e-4 | 1.122e-3 |

variant. The performance of Deepsets-variants on Burgers equation is presented in Table 10, which shows not competitive compared to MIOL. The outputs of the attention layer can be calculated by the weighted sum of attention and inputs, where the attention values have more expressive representation with extra learnable parameters $Z_0$ and inputs $a$, while the outputs of the existing pooling operation are calculated by fixed coefficient weighted sum (or operation) of inputs $a$ (in the case of avg-pooling, the weights are $\frac{1}{n_x}$). Therefore, from the perspective of the modeling operator, much of the information of interactions between pair-wise input elements is lost during those pooling operations.

Table 10: Comparisons with other set-encoder.

|  | Train grids | Test grids | | Models | |
| --- | --- | --- | --- | --- | --- |
|  | $a, u$ | $a$ | $u$ | Deepsets-variants | MIOL (ours) |
| (1) |  | 1024 | $a$ | 0.0179 | **0.0104** |
| (2) |  | 8192 | 8192 $a$ | 0.0183 | **0.0090** |
| (3) | 1024 | 1024 | 8192 | 0.0189 | **0.0106** |
| (4) |  | 512 (50% mask), | 1024 | 0.0748 | **0.0479** |

