# OpenReview forum: "Mesh-Independent Operator Learning for PDEs using Set Representations"
_ICLR.cc/2023/Conference — Submitted to ICLR 2023_

### Official Review · Reviewer_p5hq · 2022-10-19

**Confidence:** 3
**Clarity, Quality, Novelty And Reproducibility:** This method is innovative and reprodu…
**Correctness:** 3
**Technical Novelty And Significance:** 3
**Empirical Novelty And Significance:** 3
**Recommendation:** 6

**Strength And Weaknesses:**

Strength

This paper proposes a method that treats discretized functions as set-valued data without prior data structures and structurally separates dependencies on input and output meshes.

Compared with other representative models, the proposed method is evaluated on the original and extended downstream tasks. And the results show that this model is competitive in original operator learning tasks and robustly applicable in extended studies.

Weaknesses

Since this method does not require mesh, why not test some equation problems on surfaces, such as spheres?

Is it possible to solve 3D problems?

The error gap in figure 2 is vast, as the comparison methods are too simple. Can you show some more reasonable comparisons? For example, show some similar mesh-free methods.


**Summary Of The Paper:**

This paper proposes an attention-based model called the mesh-independent operator learner (MIOL) to provide proper treatments of input functions and query coordinates for the solution functions by detaching the dependence on input and output meshes. The proposed models pre-trained with benchmark datasets of operator learning are evaluated by downstream tasks to demonstrate the system's generalization abilities to varying discretization formats. The paper shows the advantages of the method with some examples.

**Summary Of The Review:**

This paper is innovative, but the presentation of results is somewhat naive. The examples shown are relatively simple. The comparison with other methods does not show the superiority of the proposed method.

---

> ### Author Response · Authors · 2022-11-18
> **Response to reviewer p5hq**
>
> We thank the reviewer's thoughtful comments with constructive feedback. Our responses are given below:
>
> 1. Why not test some equation problems on surfaces, such as spheres? / Is it possible to solve 3D problems?
>
> Since our method considers the input/output functions as set-valued data, MIOL does not require any specific data structure for the discretization of input and output.  Therefore, it is applicable to solving 3D problems. As the reviewer suggested, we added additional results (figure 4 and table 9) on learning 3D spherical shallow water which has been introduced in a very recent paper closely related to our works with publicly accessible datasets [1].
>
> 2. The error gap in figure 2 is vast, as the comparison methods are too simple. Can you show more reasonable comparisons?
>
> - First of all, since the $y$-axis of figure 2 is log-scaled, the gap seems large.
> - Although there have been many discussions about the generalization of input/output discretization raised in the PDE learning community, most existing architectures seem to be limited by their own restrictions. The main contribution of our study is to point out their own restrictions and emphasize the superiority of MIOL's generalization to arbitrary discretization format by the variety of downstream tasks. Therefore, we take standard baselines (FNO and DeepONet) for the standard benchmarks which are not trivial but generally acceptable to the PDE learning community.
> - For the case of the 2D Darcy flow problem, $-\nabla \cdot (a(x) \nabla u(x)) = f(x)$ are not time-dependent, while the mesh-free methods are originally formulated for time-stepping PDEs, i.e. $\frac{du}{dt} = F(t, u)$ [1, 2]. Therefore, we provide additional comparisons with [1, 2] on learning 3D spherical shallow water using the same evaluation measure of [1], and the result (in Table 9) shows that our method is comparable to the others.
>
> Model $\qquad\qquad$   0 < T $\leq$ 10  10 < T $\leq$ 20
>
> I-MP-PDE [2]  $\quad$  1.908e-3  $\quad$     7.240e-3
>
> DINO [1] $\qquad\quad$   1.063e-4   $\quad$   6.466e-4
>
> MIOL (ours)$\qquad$   4.524e-4    $\quad$  1.122e-3
>
> However, the quantitative results in table 9 are not a fair comparison at this moment, since [1] state that they use 16 long trajectories (i.e. 128 short trajectories) for the train set, while the accessible train data has only 8 long trajectories (i.e. 64 short trajectories). Although there are references provided for simulating the shallow water with exact system parameters, the initial states are randomly generated, therefore we didn't generate additional 8 long trajectories to meet the same size of the train set. Though only using half of the original train set, the performance of our methods is comparable to those mesh-free methods. If it is possible, we will attempt to request the full trainsets and will update our performance for a fair comparison in Table 9.
>
> [1] Y. Yin, et. al, “Continuous PDE Dynamics Forecasting with Implicit Neural Representations”, arXiv:2209.14855, 2022.
>
> [2] J. Brandstetter, et. al, “Message Passing Neural PDE Solvers”, ICLR, 2022.

---

> > ### Comment · Reviewer_p5hq · 2022-11-23
> > **Thanks**
> >
> > Since you addressed my concerns, I'm happy to increase your score.

---

> > > ### Author Response · Authors · 2022-11-23
> > > **Thank you for your reply.**
> > >
> > > We thank the reviewer for taking the time to read our response.

---

### Official Review · Reviewer_YQEY · 2022-10-26

**Confidence:** 3
**Correctness:** 3
**Technical Novelty And Significance:** 3
**Empirical Novelty And Significance:** 3
**Recommendation:** 5

**Clarity, Quality, Novelty And Reproducibility:**

The text contains only minor typographical errors (e.g., citation style on first line in Sec. 5.2). The presentation is heavy and could be improved. For example, additional visualisations could have helped demonstrate the points with less need for long descriptions in the text.

The approach is a combination of standard tools, but to my knowledge the proposed method is new.

The authors have not shared their code for review and do not mention sharing code. Implementation details are covered in main paper and appendix.

**Strength And Weaknesses:**

*Strengths*

1. The problem is topical and has received recent attention also beyond purely theoretical work. The topic area itself should be of interest for the audience of the conference.

2. The proposed approach appears sensible and described in good detail.

3. The paper is rather detailed and covers the required technical details even for someone who is familiar with the problem domain but is not an expert in the area.

*Weaknesses*

4. Even if the paper reads rather well (with only minor typos), the presentation is on the heavy side. The first 4.5 pages are describing the problem/background/preliminaries, and the paper actually starts at the bottom of page 5. The problem itself is rather simple, and could be presented in a more lightweight/approachable way, which would probably widen the impact of the work.

5. The experiments focus on rather standard benchmark problems and showing that the proposed method actually solves the problem at hand. This is valuable and provides an 'empirical' sanity check, but as a reader I would have expected this to be a first set of experiments with actual demonstrations and/or real-world problem show case data in the end.

**Summary Of The Paper:**

A paper on operator learning that does not require a specific mesh structure, but allows for querying points. Presentation a bit on the heavy side and experiments could be more demonstrative.


**Summary Of The Review:**

This is an interesting paper that considers a topical problem. The presentation is on the heavy side and the experiments only concentrate on straightforward benchmark problems.

---

> ### Author Response · Authors · 2022-11-18
> **Response to reviewer YQEY**
>
> We thank the reviewer for insightful comments with constructive feedback. All typos and grammar are corrected and our responses are given below:
>
> 1. The problem itself is rather simple, and could be presented in a more lightweight way, which would probably widen the impact of the work.
>
> Since much of the literature in the introduction was repeated in related works, we moved the related works to the appendix, and as suggested by the review, polished unnecessary sentences to lighten the presentation over the entire paper. Therefore, we can include more experimental results of more challenging results to broaden the impact of the work.
>
> 2. The experiments focus on rather standard benchmark problems ... I would have expected ... experiments with real-world problems.
>
> Although there have been many discussions about the generalization of input/output discretization raised in the PDE learning community, most existing architectures seem to be limited by their own restrictions. The main contribution of our study is to point out their own restrictions and emphasize the superiority of MIOL's generalization to arbitrary discretization format by the variety of downstream tasks. For a fair comparison, we take thorough experiments on standard benchmark problems which are not trivial but generally acceptable to the PDE learning community. To widen the impact of our work, we added additional results (figure 4 and Table 9) on learning 3D spherical shallow water which has been introduced in a very recent paper closely related to our works with publicly accessible datasets [1]. Although it is not a real-world problem, this problem is definitely a more challenging problem compared to the standard benchmark problems, where the discretized points are placed on the 3D spherical surface. As the reviewer mentioned (and also discussed in our section 7. Limitation), applications for realistic scenarios were not included, and we leave it for future works.
>
> 3. The authors have not shared their code for review and do not mention sharing code. Implementation details are covered in main paper and appendix.
>
> As the reviewer mentioned, the implementation details are described in the manuscript, and we will make the code released on Github in near future.
>
> [1] Y. Yin, et. al, “Continuous PDE Dynamics Forecasting with Implicit Neural Representations”, arXiv:2209.14855, 2022.

---

### Official Review · Reviewer_kSWf · 2022-10-27

**Confidence:** 3
**Correctness:** 3
**Technical Novelty And Significance:** 3
**Empirical Novelty And Significance:** 3
**Recommendation:** 5

**Clarity, Quality, Novelty And Reproducibility:**

The paper discusses an important topic in the context of operator learning models, but I think the paper lacks in the quality of the exposition. There are wrong linking words and logical phrases in the text (check again the grammar!) and redundancies in the sentences. Some statements are vague (see below). Since the proposed model is inspired by several existing architectures, some more background information would be useful for clarity. Figure 3 seems rushed to finish.
In the experiments, Task 4 is claimed to be  a novelty compared to existing models, so it would be interesting to go more in depth with it (not only in the appendix).

Some more detailed comments/questions:

  – Some function spaces (\mathcal{A}, \mathcal{F}) are not specified

– phrases like “ to capture the interactions between the elements” are vague

– how is (2) evaluated?

– In (5), the approximation with the integral is not clear to me. Could you please add details?

– related to Task 4: with MIOL, can I then use any discretization I want in the input space, and get my approx. solution be evaluated on any discretization in the output function, independently of the discretization chosen in the input space? And if yes, what is the error I will encounter? These are I think the main questions that should be addressed.



**Strength And Weaknesses:**

I think the paper has his strength in raising an important question regarding discretization-flexibility w.r.t input/output functions  for operator learning models. The paper proposes an architecture that combines properties of existing operators (such as neural operators and set transformers) to address this issue and shows in experiments its performance. However, I think there are some weaknesses in the presentation of the paper:

–  Some more background on the used architectured would be appreciated

– The paper lacks a discussion about the error bounds w.r.t the discretization chosen

– The authors should go again through the text since it is at times redundant  and imprecise in the linking words and logical phrases

– In the experiments, the figures need to be polished. In particular Figure 3 (fontsize of colorbar ticks too small, x and y axes missing....)


**Summary Of The Paper:**

Operator learning models for PDEs learn the parameter-to-solution mapping of families of partial differential equations. Since they map functions into functions, their approximation should be independent of the discretization of the respective function spaces. This is not always the case for the operator learning networks proposed until now: in practice they often evaluate the functions on finite discretizations and are not able to evaluate the inputs and outputs on different meshes, or to treat training and testing sets with different discretization meshes. The paper raises these issues and develops MIOL- a mesh-independent attentional-architecture that separates dependencies on input and output meshes. It compares its performance with the existing models on several tasks.

**Summary Of The Review:**

I think the paper raises an important issue in the context of operator learning models. However, the paper needs some polishing of the form of the exposition, a more in depth discussion regarding discretization and error bounds, and some Figure polishing.

---

> ### Author Response · Authors · 2022-11-18
> **Response to reviewer kSWf (part 1)**
>
> We thank the reviewer's insightful comments with constructive feedback. As the reviewer suggested, all typos and sentences are corrected and our responses are given below:
>
> 1. How is (2) evaluated? / lacks a discussion about the error bounds / Related to task 4: ... and if yes, what is the error I will encounter?
>
> As the reviewer suggested, we have updated the calculation of the empirical test errors and discussion about the error bounds related to task 4. First of all, our model MIOL can be evaluated on any discretization in the output function, independently from the discretization chosen in the input domain. The followings are the answer to the questions, also found in section 2 of our revised manuscript (problem definition, page 3).
> For a fair comparison, MIOL is trained by the same training procedure as existing neural operators, where the originally given input and output pairs have the same discretization (equation (1)). After that, we attempt to validate the models with the following test error (equation (2)),
>
> $$E_{a \sim \mu}E_{X, Y} [\mathcal{L} ([ \mathcal{G} (a) ] (y), u(y)) ] = E_{a \sim \mu} \int_{\Omega_x} \int_{\Omega_y} \lVert u(y) - [\mathcal{G} (a|_{X})] (y) \rVert^2 dy d\nu(X),$$
>
> where $X$ and $Y$ are arbitrary discretizations of the input and output domains. In task 4, instead of calculating the exact test error, we calculate respective empirical test errors corresponding to masking ratio $p$,
>
> $$\frac{1}{N} \sum_{i=1}^N \lVert u_i(Y) - [\mathcal{G} (a_i^{(p)} ) ] (Y) \rVert^2.$$
>
> where $a_i^{(p)} = a_i |_{X^{(p)}}$ is randomly masked discretization input $a_i$ with masking ratio $p$, and the model is evaluated at all given discretized points of $Y$ (using all given points of $Y$ is not necessary). The empirical test error can be bounded by a sum of approximation error and discretization error of discrepancy from the discretizations of input functions,
>
> $$ \lVert u_i(Y) - [ \mathcal{G} (a_i^{(p)} ) ] (Y) \rVert \leq  \lVert u_i(Y) - [ \mathcal{G} (a_i) ] (Y)  \rVert + \lVert [ \mathcal{G} (a_i) ] (Y) - [ \mathcal{G} ( a_i^{(p)} )  ] (Y) \rVert.$$
>
> The approximation error is expected to be sufficiently small by training procedure (by equation (1)). If we consider the input functions as set representation, i.e., $a_i^{(p)} \subset a_i$, there exists a threshold of masking ratio $p$ where the discretization error can be expected to be sufficiently small by processing them with efficient permutation-invariant set encoder whose outputs are independent to cardinality and locations of input discretization [1, 2]. In section 5.2 (Figure 5 and Table 3), when the masking ratio is lower than about 50%, the errors are consistently low, since the efficient set encoder can aggregate sufficient information from the subsampled input $a_i^{(p)}$ to make small discretization error.
>
> 2. Some more background on the used architecture would be appreciated / In (5), the approximation with the integral is not clear to me.
>
> In Appendix A. 3 (Detail explanation of equation 7), we provide the explanation for the approximation with the integral for the cross-attention mechanism, where the softmax for the attention matrix is ignored for simplicity. From the explanation, the discretization of input is {$x_1, ..., x_{n_x}$} (as key and value vectors), and it can be changed to the discretization of output {$y_1, ..., y_{n_y}$} (as query vectors) with cardinality changed from $n_x$ to $n_y$. Using this mechanism, we can detach the dependences of discretization formats of input and output from the processor, by encoding arbitrary discretization {$x_1, ..., x_{n_x}$} to a fixed size ($n_z$) of learnable latent set vectors, and decoding the latent set vectors to output arbitrary discretization {$y_1, ..., y_{n_y}$}. The discretization number is varied as $n_x$ (arbitrary) $\rightarrow$ $n_z$ (fixed) $\rightarrow$ $n_y$ (arbitrary).
>
> 3. In the experiments, the figures need to be polished. In particular figure 3 (...).
>
> We polished figure 3 with the larger font size for ticks. But the $x, y$ axes are not included because they make the figure messy. Note that all the $x, y$ axes range from 0 to 1.
>
> 4. Some function spaces are not specified.
>
> Since the descriptions in the introduction are for a brief explanation that the approximated model $\mathcal{G}: \mathcal{A} \rightarrow \mathcal{U}$ can be considered as a mapping between the Banach spaces $\mathcal{A}$ and $\mathcal{U}$, we updated the specifications for the function spaces in Section 2 (problem definition), instead of updating the descriptions in the introduction.

---

> ### Author Response · Authors · 2022-11-18
> **Response to reviewer kSWf (part 2)**
>
> 5. Phrases like "to capture the interactions between the elements" are vague.
>
> Since modeling PDE can be interpreted as discovering the governing laws of interactions (or kernel values of operators) between infinitesimal segments, the phrases mean that the approximated model should have the ability to capture the interactions or kernel values between sampled elements from the function. For this reason, it is reasonable that the architectures for operator learning are usually composed of kernel integral operations, and we approximated them with attentional modules which have been discussed as efficient tools to express those pair-wise interactions. Also, the phrases are common expressions in works of literature on set representations [1, 2], attention [3, 4], and kernel integration [5, 6].
>
> [1] M. Zaheer. Et. al, “Deep Sets”, NeurIPS, 2017.
>
> [2] E. Wagstaff, et. al, “Universal Approximation of Functions on Sets”, Journal of Machine Learning Research, 2022.
>
> [3] H. Kin, et. al, “Attentive Neural Processes”, ICLR, 2019.
>
> [4] S. Cao, et. al, “Choose a Transformer: Fourier or Galerkin”, NeurIPS, 2021.
>
> [5] Z. Li, et. al, “Multipole Graph Neural Operator for Parametric Partial Differential Equations”, NeurIPS, 2020.
>
> [6] N. Kovachki, et. al, “Neural Operator: Learning Maps Between Function Spaces with Applications to PDEs”, arXiv:2108.08481, 2021.

---

### Author Response · Authors · 2022-11-17
**General response**

We would like to appreciate all the reviewers for their constructive comments which have led to the revision and we believe without a doubt have improved the quality of the manuscript. The revised manuscript has been uploaded. We have addressed the individual reviewer's comments below respectively, and here the summary of the major changes is as follows:
1. We moved related works to the appendix since much of the literature in the introduction was repeated in related works.

2. In section. 2 (problem definition), we added the discussion of the evaluation of empirical test error and error bounds (related to task 4 in experiments), and regarding this discussion, we moved the "performances according to input masking ratios" from the appendix to the section. 5.2 (experimental results).

3. We conducted the experiments for the 3D spherical shallow water dataset [1] as presented in figure 4 and Table 9 (in the appendix).

[1] Y. Yin, et. al, "Continuous PDE Dynamics Forecasting with Implicit Neural Representations", arXiv:2209.14855, 2022.

---

### Decision · Program_Chairs · 2023-01-20

**Decision:**

Reject

**Justification For Why Not Higher Score:**

see above

**Justification For Why Not Lower Score:**

N/A

**Metareview: Summary, Strengths And Weaknesses:**

All reviewers agreed (in the offline discussion) that the paper is an interesting contribution, but suffers from poor presentation and some conceptual weaknesses (details in the reviews and discussion below). The reviewers stressed that the authors should feel encouraged to resubmit the paper after these technical issues have been resolved.

**Summary Of Ac-Reviewer Meeting:**

Because the reviewers live in different time zones, and have busy schedules, it was not possible to schedule an in-person meeting at short notice. There was an email thread instead. The results are above.